# Protective Effect of *Lactiplantibacillus plantarum* 1201 Combined with Galactooligosaccharide on Carbon Tetrachloride-Induced Acute Liver Injury in Mice

**DOI:** 10.3390/nu13124441

**Published:** 2021-12-12

**Authors:** Zhongyue Ren, Yalan Huo, Qimeng Zhang, Shufang Chen, Huihui Lv, Lingling Peng, Hua Wei, Cuixiang Wan

**Affiliations:** 1State Key Laboratory of Food Science and Technology, Nanchang University, Nanchang 330047, China; zhongyueR666@163.com (Z.R.); tan020624@163.com (Q.Z.); shufangchen2021@163.com (S.C.); ncuspylvhuihui@163.com (H.L.); penglingling6@163.com (L.P.); weihua@ncu.edu.cn (H.W.); 2Department of Medicinal Chemistry and Molecular Pharmacology, College of Pharmacy, Purdue University, 575 W Stadium Ave, West Lafayette, IN 47907, USA; huo15@purdue.edu; 3Sino-German Joint Research Institute, Nanchang University, Nanchang 330047, China

**Keywords:** *Lactobacillus plantarum* 1201, galactooligosaccharides, synbiotic, acute liver injury, intestinal flora

## Abstract

Acute liver injury (ALI) has a high mortality rate of approximately 20–40%, and it is imperative to find complementary and alternative drugs for treating ALI. A carbon tetrachloride (CCl_4_)-induced ALI mouse model was established to explore whether dietary intervention can alleviate ALI in mice. Intestinal flora, intestinal integrity, biomarkers of hepatic function, systemic inflammation, autophagy, and apoptosis signals were detected through a real-time PCR, hematoxylin-eosin staining, 16S rRNA gene sequencing, and so on. The results showed that *Lactiplantibacillus plantarum* 1201 had a strongly antioxidant ability, and galactooligosaccharide (GOS) could boost its growth. Based on these findings, the combination of *L. plantarum* 1201 and GOS, the synbiotic, was applied to prevent CCl_4_-induced ALI in mice. The current research proved that GOS promoted the intestinal colonization of *L. plantarum* 1201, and the synbiotic improved the antioxidant capacity of the host, regulated the intestinal flora, repaired the intestinal barrier, inhibited the activation of the MAPK/NF-κB pathway, and then inhibited the apoptosis and autophagy pathways, relieving inflammation and liver oxidation; thereby, the ALI of mice was alleviated. These results suggest that synbiotics may become a new research direction for liver-protecting drugs.

## 1. Introduction

The liver plays many important roles in the body, such as the detoxification of chemicals, including drugs and environmental contaminants, which, in turn, damage the liver [1]. The liver is also able to maintain metabolic homeostasis [2]. Acute liver injury (ALI) is a life-threatening clinical syndrome characterized by rapid loss and abnormal liver cell function in patients with or without previous liver disease [3]. The molecular process of ALI is thought to involve complex interactions between oxidative stress, apoptosis, autophagy, and necrosis [4,5]. In the early stages of ALI, the initial purpose of the liver is to induce the generation of reactive oxygen species (ROS) in hepatocytes and endoplasmic reticulum stress, thereby triggering hepatocyte apoptosis and necrosis. If the inactivated liver cells are not cleared quickly by the phagocytic cells, such as the phagocytic Kupffer cells, the membranes of these liver cells will break down, leading to secondary necrosis [6,7,8]. Therefore, a focus on reducing the oxidative damage of the liver is of great significance for the prevention of ALI.

In recent years, with the introduction of the concept of the gut–liver axis, some researchers have boldly proposed that ALI is also related to intestinal flora. When the normal liver physiology is destroyed, certain components of bacteria, such as lipopolysaccharide (LPS), will enter the liver from the intestinal cavity through the portal vein, causing excessive inflammation, which further damages the tissues and aggravates the original liver disease [9,10,11]. Under normal circumstances, local and systemic protection and mechanisms will prevent most bacteria and their products from penetrating the intestinal wall, including intestinal colonization resistance [12,13]. Intestinal damage can also aggravate liver damage. Arsenic exposure induces intestinal barrier damage and consequently leads to inflammation and pyroptosis of the liver in ducks [14]. Intestinal *Clostridioides difficile* can initiate and aggravate liver injury through the occurrence of inflammation and damage to the hepatocytes [15]. Therefore, this research focused on the intestinal flora, and compared with previous findings, we further considered the effect of adjusting the intestinal microecological balance on the prevention of ALI.

Probiotics are defined as viable microorganisms, sufficient amounts of which reach the intestine in an active state and thus exert positive health effects [16]. A prebiotic is a selectively fermented ingredient that allows specific changes, both in the composition and/or activity in the gastrointestinal microflora that confers benefits upon the host’s wellbeing and health, whereas synergistic combinations of pro- and prebiotics are called synbiotics [17]. Probiotics or prebiotics have been proven to be safe and effective in regulating intestinal flora. Probiotics can colonize in the intestinal tract, supplement beneficial flora, maintain the balance of intestinal bacteria, and competitively inhibit the colonization of pathogenic bacteria [18]. Previous studies have shown the benefit of probiotics for the concentrations of liver enzymes in the serum, liver steatosis, lipid profiles, and liver stiffness [19,20]. In addition, it has been found that *Lactobacillus* or *Clostridium butyricum* supplements can reduce the ALI caused by CCl_4_ [21,22]. Galactooligosaccharide (GOS) is a prebiotic which is the only oligosaccharide derived from animal milk that can be used by the eight beneficial bacteria in the human intestine [23]. Additionally, it is an excellent source of nutrients and effective proliferation factors for beneficial bacteria such as bifidobacteria and *Lactobacillus* in the human intestine [24]. In addition, GOS has proved that it can regulate intestinal flora [25] and protect against DSS-induced intestinal injury through inhibiting the NF-κB pathway [26]. However, some researchers found that when probiotics were taken alone, exogenous probiotics and intestinal microbes led to competition for nutrition and space, which increased the instability of the gut microbiota [27]. The appropriate supplementation of nutritional substrates as prebiotics for the gut microbial community could lower such competition and therefore enhance the colonization of probiotic strains [28]. Therefore, we wanted to know whether the synbiotic (the combination of probiotics and prebiotics) had a better effect on the prevention of ALI.

In this study, *Lactiplantibacillus plantarum* 1201 was isolated from Fuzhou preserved pickles. We found that *L. plantarum* 1201 had an excellent antioxidant capacity, which meant that *L. plantarum* 1201 might reduce oxidative damage to inhibit the ALI caused by CCl_4._ Additionally, GOS could promote the growth of *L. plantarum* 1201. The aim of the present study was to investigate and compare the hepatoprotective effects of *L. plantarum* 1201, GOS, and a synbiotic (the combination of *L. plantarum* 1201 and GOS) using a CCl_4_-induced ALI murine model, to provide a new perspective for liver protection.

## 2. Materials and Methods

### 2.1. The Lactobacillus Strain and Culture Conditions

*L. plantarum* 1201 was cultured under anaerobic conditions at 37 °C in sterile deMan, Rogosa, Sharpe broth (Beijing Solarbio Science & Technology, Co. Ltd., Beijing, China). Subsequently, cells were harvested, centrifuged at 6000 rpm for 5 min at 25 °C, and washed in sterile 1 × PBS. *L. plantarum* 1201 isolated from Fuzhou Preserved Pickles is deposited in the China Center for Type Culture Collection with the collection number CCTCC M 2021050.

### 2.2. Antioxidant Ability of the Lactobacillus Strain

#### 2.2.1. Preparation Culture Supernatant and the Intracellular Extract

We set up four groups to measure the antioxidant capacity of *L. plantarum* 1201, namely a cell-free supernatants group (CFS), a cell suspension group (CS), a cell free extracts group (CFE), and a positive control group (PC). *Lacticaseibacillus rhamnosus* GG has been proven by previous studies to have a good antioxidant capacity [29], and some researchers have used LGG as a control to evaluate the antioxidant capacity of probiotics [30]. So, in this study, LGG was used as a control strain to evaluate the antioxidant capacity of *L. plantarum* 1201.

A total of 10 mL MRS was used to incubate *L. plantarum* 1201 at 37 °C for 24 h, then we transferred aliquots of the culture to 2 mL polypropylene tubes, and centrifuged at 12,000 rpm for 15 min at 4 °C. Next, 1 M NaOH was used to neutralize (pH 7.0) the supernatant and the mixture was heated at 95 °C for 5 min. According to the previous research [31], ultrapure water was used to wash the cell pellet and suspend it, then the cell pellet was sonicated (Unique USC 700) at 40 kHz for 15 min, with 5 intervals of 1 min in an ice bath [31]. Cellular debris was removed by centrifugation at 12,000 rpm for 15 min at 4 °C. Finally, we obtained the intracellular extracts.

#### 2.2.2. DPPH Radical Scavenging Assay

A DPPH radical solution (0.2 mmol/L) in 95% ethanol was prepared. A total of 1 mL of DPPH in ethanol was added to 1 mL of the sample (culture supernatant or intracellular extract), well vortexed, and incubated for 30 min in a dark room at room temperature. Subsequently, the mixture was centrifuged at 7200 rpm for 10 min. The absorbance of the supernatant was measured with a spectrophotometer at 517 nm. An equal volume of distilled water, was used for the control group and ethanol was used as a blank. The antioxidant activity was expressed as a percentage of DPPH activity calculated as:$$\mathrm{DPPH}\;\mathrm{activity}\;\left(\%\right)=\frac{\mathrm{Absorbance}\;\mathrm{of}\;\mathrm{blank}\;-\;\mathrm{absorbance}\;\mathrm{of}\;\mathrm{sample}}{\mathrm{Absorbance}\;\mathrm{of}\;\mathrm{blank}}\times 100$$

#### 2.2.3. Reducing Power Assay

The 0.5 mL of the 2 mM phosphate buffer (pH6.6) and 1% of Potassium ferricyanide (0.5 mL) were added to 0.5 mL of the sample and kept at 50 °C for 20 min. A total of 10% trichloroacetic acid (0.5 mL) was added to the reaction mixture and spun at 3000 rpm for 10 min. Distilled water (1 mL) and 1% ferric chloride (1 mL) was added to incubate for 10 min, then the absorbance of the mixture was measured at 700 nm. We used an equal volume of the phosphate buffer in the blank group to replace the sample solution [32].
$$\mathrm{Reducing}\;\mathrm{power}\;\left(\%\right)\;=\frac{\mathrm{Absorbance}\;\mathrm{of}\;\mathrm{blank}\;-\;\mathrm{absorbance}\;\mathrm{of}\;\mathrm{sample}}{\mathrm{Absorbance}\;\mathrm{of}\;\mathrm{blank}}\times 100$$

#### 2.2.4. Hydroxyl Free Radical Scavenging Activity Assay

The reaction mixture containing 1.0 mL of PBS (20 mM, pH 7.4), 0.5 mL of 1,10-phenanthroline (2.5 mM, Sigma-Aldrich, Shanghai, China), 0.5 mL of FeSO_4_ (2.5 mM), 0.5 mL of H_2_O_2_ (2.5 mM), and 0.5 mL of the samples was incubated at 37 °C for 60 min. The absorbance of the mixture was measured at 536 nm after centrifugation (3000 rpm, 10 min). LGG was used to evaluate the ability of *L. plantarum* 1201.
$$\mathrm{Hydroxyl}\;\mathrm{radical}\;\mathrm{scavenging}\;\mathrm{activity}\;\left(\%\right)=\frac{\mathrm{Absorbance}\;\mathrm{of}\;\mathrm{sample}\;-\;\mathrm{absorbance}\;\mathrm{of}\;\mathrm{control}}{\mathrm{Absorbance}\;\mathrm{of}\;\mathrm{blank}\;-\;\mathrm{absorbance}\;\mathrm{of}\;\mathrm{control}}\times 100$$

### 2.3. Growth Experiments

The medium was treated with 20, 30, and 40 g/L GOS, and the MRS medium was set as the control group. A total of 100 μL of *L. plantarum* 1201 was inoculated with 10 mL of the medium. The optical density at 630 nm was measured every 3 h for 24 h in an anaerobic cabinet to monitor the growth of *L. plantarum* 1201 (36 °C ± 1 °C; 80% N_2_, 10% CO_2_, 10% H_2_). Three independent experiments, each performed in triplicate, were run.

### 2.4. Animals

Six-week-old male specific pathogen-free C57BL/6N mice (Beijing Vital River Laboratory Animal Technology, Co. Ltd., Beijing, China) were housed in Nanchang Royo Biotech, Co. Ltd., Nanchang, China. under standard conditions with a light/dark cycle of 12 h. All mice were provided with ad libitum access to food and water. All experimental procedures were in accordance with the guidelines of the Institutional Animal Care and Use Committee of Nanchang Royo Biotech, Co. Ltd.

### 2.5. Experimental Groups

Mice were randomly assigned to 5 groups (ND, MD, LD, PD, LP) with 10 mice in each group. (1) the ND group was given PBS for 2 weeks, and 1 h after the last gavage, 2 μL/g peanut oil solution was injected intraperitoneally; (2) mice in the MD group were given PBS for 2 weeks, and 1 h after the last gavage, 2 μL/g 50% CCl_4_ peanut oil solution was injected intraperitoneally; (3) mice in the LD group were administered intragastrically with 3 × 10^8^ cfu/mL *L. plantarum* 1201 for 2 weeks, and 1 h after the last gavage, 2 μL/g 50% CCl_4_ peanut oil solution was injected intraperitoneally; (4) mice in the PD group were given PBS for 1 week first, then were administered 100 μL GOS intragastrically at a concentration of 0.5 g/kg for 1 week. Moreover, 1 h after the last gavage, 2 μL/g 50% CCl_4_ peanut oil solution was injected intraperitoneally; (5) mice in the LP group were given 3 × 10^8^ cfu/mL *L. plantarum* 1201 for 2 weeks, and 100 μL GOS at a concentration of 0.5 g/kg for 1 week. Moreover, 1 h after the last gavage, 2 μL/g 50% CCl_4_ peanut oil solution was injected intraperitoneally.

A total of 16 h after intraperitoneal injection, the mice were euthanized with ether. The mouse serum, liver, colon, colon contents, ileum, and cecum contents were separately collected in a sterile centrifuge tube and stored at −80 °C for subsequent experiments. The plasma was placed in a refrigerator at 4 °C for 1 night and centrifuged at 3000 rpm for 10 min, then the upper layer of serum was taken for storage.

### 2.6. Organ Index Determination

The mice were weighed with an electronic scale (Kaifeng Group, Co. Ltd., Yongkang, China). Then the neck was severed, and the liver tissue was dissected and separated. Connective tissues such as fat and fascia were removed. Then they were rinsed with normal saline, and after absorbent paper was used to absorb the surface water, the organs were weighed and recorded. The liver index was calculated as the liver weight divided by the body weight.

### 2.7. Aminotransferase Measurement

Serum alanine aminotransferase (ALT) and apartate transaminase (AST) were measured with commercial Assay Kits (Jiancheng Bioengineering, Nanjing, China).

### 2.8. Examination of Oxidation Markers

Liver samples (100 mg) were homogenized in 900 μL of saline at 4 °C and then centrifuged at 4000 rpm at 4 °C for 10 min. According to the manufacturer’s instructions, superoxide dismutase (SOD), glutathione (GSH), and malondialdehyde (MDA) in the liver tissue were measured using commercial test kits (Jiancheng Bioengineering, Nanjing, China).

### 2.9. Hematoxylin and Eosin Staining

Fresh colon and liver tissues were collected and soaked in 10% formalin tissue fixative solution. After the alcohol was dehydrated, the tissue blocks were sequentially transparent, waxed, encased, sliced, and baked in turn. The conventional dewaxing process was to sequentially put paraffin sections into xylene I (10 min)—xylene II (10 min)—ethanol I (5 min)—ethanol II (5 min)—95% alcohol (3 min)—90% alcohol (3 min)—80% alcohol (2 min)—70% alcohol (2 min). Next, the slices were soaked in distilled water and washed for 2 min and stained with hematoxylin and eosin (H&E), then the slices were put into anhydrous ethanol I for 5 min, anhydrous ethanol II for 5 min, xylene I for 5 min, and xylene II for 5 min, and dried to a neutral gum sealing piece before microscopy.

### 2.10. DNA Extraction from the Mouse Cecal Contents and 16S rRNA Gene Sequencing

The genomic DNA of mouse cecal contents was extracted, and the purity and concentration of DNA were detected. High-throughput sequencing was conducted by Biomarker Tech (Beijing, China). Clean Data were obtained by stitching, filtering, and removing the chimera of the original Data. Then the clustering and species classification of the Operational Taxonomic Units (OTUs) were analyzed based on the available data. Based on the results of the OTU cluster analysis, a multiple diversity index analysis and sequencing depth detection were conducted for OTUs. Based on the taxonomic information, the community structure was statistically analyzed at each taxonomic level.

The representative sequences of OTUs were annotated with the Greengenes database. A Venn diagram was drawn based on the OTU distribution. Based on the analysis of the species composition, the community structure component map, the community Heatmap, and the UPGMA clustering tree based on Unweighted Unifrac and weighted Unifrac distance were obtained. The difference analysis, PCA analysis, PCOA analysis, and NMDS analysis from the Beta diversity were performed using a *t*-test, a Wilcox Rank Sum test, and a Tukey test. The LEfSe analysis was based on taxonomic composition, and the linear discriminant analysis (LDA) was performed on the samples according to the different grouping conditions to find out the communities or species with significant differences in sample classification.

### 2.11. Gene Expression

High quality RNA was isolated from the frozen colon and liver tissue using the TaKaRa RNA extraction kit (Takara, Otsu, Japan), and then used for cDNA synthesis with a transcriptor cDNA kit (Takara, Otsu, Japan).

The mRNA levels of the pro-inflammatory factors (IFN-γ, IL-1β, TNF-α, IL-6), anti-inflammatory factors (IL-10, IL-22, TGF-β), fibronectin 1 (FN1), Chemokines (CCL4, CCL5), Phospho-extracellular regulated protein kinases (p-ERK), c-Jun N-terminal kinase (JNK), and B-cell lymphoma-2 (Bcl-2) in the liver tissue were measured using a real-time PCR. The mRNA levels of the pro-inflammatory factors (IFN-γ, IL-1β, TNF-α, IL-6), anti-inflammatory factors (IL-10, IL-22, TGF-β), Chemokines (CCL4, CCL5), intestinal permeability protein-related gene, (clan3, ocln, ZO-1), p-ERK, JNK, and Bcl-2 in the colon tissue were measured using a real-time PCR. The PCR reaction procedure had three steps: 5 µL SYBR Green Mix, 0.8 µL primer (10 µM), 1 µL cDNA (1000 ng/µL) plus a ddH_2_O supplement in the 10 µL system. The reaction conditions involved preheating at 95 °C for 30 s; a cycling stage: denaturation at 95 °C for 5 s; 59 °C return for 1 min; 72 °C extension for 30 s, 40 cycles; and a melting stage: 65 °C 5 s, 95 °C, 5 s. The primer sequence information is shown in the Appendix A.

### 2.12. Enzyme-Linked Immunosorbent Assay

According to the kit instructions (Shanghai C-reagent Biotechnology, Co. Ltd., Shanghai, China), the double antibody sandwich method was used to determine the protein expression levels of mouse Caspase-3, BAX, Bcl-2, Beclin1, Atg5, and microtubule-associated protein 1 light chain 3 (LC3-II) in serum.

### 2.13. Statistical Analysis

All values are expressed as mean ± standard error of the mean (SEM). The Graph Pad Prism 6 (GraphPad, Inc. La Jolla, CA, USA) was employed for statistical analysis and graph preparation. A one-way ANOVA was used for multiple comparisons. Statistical significance was considered at *p* < 0.05.

## 3. Results

### 3.1. GOS Promoted the Growth of L. plantarum 1201 with High-Quality Antioxidant Power

*L. plantarum* 1201 growth accelerated after 6 h. After 21 h, the cell number reached its highest level. At this time, the OD 630 nm value of all the bacterial suspensions reached levels above 1.0, and the proliferation and death of the strain gradually balanced. The growth of *L. plantarum* 1201 was affected by the addition of GOS. As shown in Figure 1A, the growth curves of *L. plantarum* 1201 in 20, 30, and 40 g/L of the GOS-added media were all higher than that of the control, with 20 g/L of the GOS medium having the best effect, which is consistent with previous reports [33,34].

The different components of *L. plantarum* 1201 showed significant differences in relation to the scavenging rate of DPPH free radicals (Figure 1B). LGG has been proven to have a strong antioxidant capacity [35]. Compared with LGG, *L. plantarum* 1201 had a similar effect on DPPH free radical scavenging. Their scavenging rates were 83.0% and 82.1% in the CFS group, while the scavenging abilities of the CS and CFE groups were both below 20%. This shows that *L. plantarum* 1201 has the active substance related to the DPPH free radical scavenging ability mainly in extracellular metabolites.

The hydrogen peroxide activity of *L. plantarum* 1201 is shown in Figure 1C. In the CS group, the scavenging rates of *L. plantarum* 1201 and LGG on hydroxyl free radicals were 82.3% and 91.5%, respectively, indicating that the living bacteria had strong antioxidant activity. The scavenging rate of the cell extract (CFE) of *L. plantarum* 1201 on hydroxyl free radicals was only 7.7%, nearly half of LGG, indicating that its intracellular products had a weak scavenging ability in relation to hydroxyl free radicals.

As shown in Figure 1D, the absorbance of the *L. plantarum* 1201 CFS group was 0.69, while the absorbances of the CS group and the CFE group were 0.27 and 0.23, respectively. The results of LGG were consistent with 1201, the absorbance of the CFS group was 0.81, and the absorbance of the CS group and the CFE group were, respectively, 0.25 and 0.23. This shows that the active substance related to the reducing ability of *L. plantarum* 1201 and LGG may be the secretion of extracellular metabolism products.

### 3.2. Synbiotics Alleviated the Dysfunction and Pathological Damage of the Liver in CCl_4_-Induced ALI Mice

As shown in Figure 2B, CCl_4_ treatment caused an increase in the organ index (*p* < 0.001). However, compared with MD, the levels of the organ index were significantly reduced by *L. plantarum* 1201, GOS, and synbiotic pretreatment (*p* < 0.01).

To detect whether *L. plantarum* 1201 and GOS could reduce the liver dysfunction induced by CCl_4_, serum enzymes were analyzed. As shown in Figure 2C,D, CCl_4_ treatment increased the activities of ALT and AST (*p* < 0.001) in the serum. However, *L. plantarum* 1201, GOS, and synbiotic pretreatment reduced the activities of ALT and AST. Among them, GOS and the synbiotics had significant effects (*p* < 0.01).

We also used H&E staining of the liver tissues to analyze the histological changes in these different groups. Through the observation method of the H&E stained sections in the liver, we found severe hepatocellular necrosis and inflammatory cell infiltration, especially around the central veins in the model group [36]. However, the liver lesions of the LD, PD, and LP groups were drastically attenuated, with synbiotic pretreatment showing the most significant relief (Figure 2E).

### 3.3. Synbiotics Reduced Oxidative Stress and Relieved Liver Inflammation in CCl_4_-Induced ALI Mice

Oxidative stress is a key part of the pathogenesis of ALI. We examined the SOD activities and the contents of MDA and GSH in liver homogenates, the key markers of oxidation. We found that the CCl_4_ challenge decreased the SOD activity by 39% and reduced the content of GSH by 25.98%, while it increased the content of MDA in the liver by 44.89%. However, *L. plantarum* 1201, GOS, and synbiotic treatment had the opposite effect, as they significantly increased the SOD activity and the content of GSH, while reducing the content of MDA (*p* < 0.01, Figure 3A–C).

To investigate whether *L. plantarum* 1201, GOS, and synbiotics could attenuate liver inflammation, we detected the expression levels of some pro-inflammatory cytokines (IL-6, IL-1β, TNF-α, IFN-γ), anti-inflammatory factors (IL-10, IL-22, TGF-β), and chemokines (CCL4, CCL5) in liver tissues. The CCl_4_ challenge decreased the expression level of IL-10 in the liver by 53.27%, while it drastically increased the mRNA level of IL-22, IL-1β, TNF-α, IL-6, IFN-γ, TGF-β, CCL4, and CCL5 by 200% or more. However, with *L. plantarum* 1201, GOS, and synbiotic pretreatment, the LD, PD, and LP groups yielded similar results to the control group (Figure 3D,E). Furthermore, the mRNA level of Bcl-2 was also noticeably decreased in the MD group compared with the ND and diet intervention groups, whereas the mRNA expression level of FN-1 was significantly elevated with diet intervention (Figure 3F,G).

### 3.4. Synbiotics Alleviated Intestinal Flora Disturbance in CCl_4_-Induced ALI Mice

Due to the presence of the liver–gut axis, we speculated that the gut microbiota would change with liver damage. We used both α and β diversity analyses to assess the differences between the groups in the diversity of intestinal flora. A principal coordinates (PCoA) analysis was used to analyze the β-diversity among groups. The distribution of the intestinal flora in the MD group was quite different from that in the ND group. However, with *L. plantarum* 1201, GOS, and synbiotic treatment, the microbial composition was similar to the ND group (Figure 4A,B). Moreover, the species rarefaction curves tended to be relatively flat, illustrating that the sequencing quantity and depth were qualified, and we could carry out subsequent analysis (Figure 4C). The diversity (Shannon) and community abundance index (Chao) were used to reflect the α diversity of the intestinal microbial population [14]. Compared with those of the ND group, the Shannon and Chao indexes were significantly decreased in the MD group, and the dietary intervention groups yielded similar indexes to the control group (Figure 4D,E), which indicated that CCl_4_ significantly altered the richness and diversity of the intestinal microbiota in mice, and *L. plantarum* 1201, GOS, and synbiotic showed preventive effects.

We further analyzed the changes in the microbiota at the phylum and family levels. At the phylum level (Figure 4H), a Metastats analysis showed that the CCl_4_ challenge significantly decreased the abundance of *Firmicutes* while increasing the abundance of *Bacteroidetes*, *Proteobacteria*, and *Actinobacteria.* Additionally, *L. plantarum* 1201, GOS, and synbiotic treatment prevented these changes (Figure 4F). At the family level, the relative abundances of *Aeromonadaceae*, *Akkermansiaceae*, *Bacteroidaceae*, and *Lachnospiraceae* were more abundant in the CCl_4_-treated group than in the control group, while the relative abundances of *Lactobacillaceae* and *Christensenellaceae* were more abundant in the control and dietary intervention groups (Figure 4G–I).

Through the family-level flora correlation network, we found that *Lactobacillaceae* and *Christensenellaceae* were positively correlated (Figure 4J). This is consistent with the changes in the flora abundance at the family level and is also consistent with the previous literature [37].

### 3.5. Synbiotics Relieved Intestinal Inflammation in CCl4-Induced ALI Mice

Flora disorders are usually accompanied by inflammation. Therefore, we tested the expression of inflammatory factors in the intestinal tissue. Similar to the inflammation of the liver, the CCl4 challenge decreased the expression level of IL-10 in the colon, while it drastically increased the mRNA expression level of IFN-γ, IL-1β, TNF-α, IL-6, IL-22, TGF -β, CCL4, and CCL5. The mRNA expression level in the diet intervention groups was close to normal level (Figure 5A,B). Additionally, a significant difference in the mRNA expression level of Bcl-2 occurred in CCl_4_-treated group, while it was similar between the control and dietary intervention groups (Figure 5C).

Intestinal tight junctions (TJs) have been shown to be associated with intestinal barrier integrity [38]. To test whether there was intestinal barrier destruction in the CCl_4_-induced ALI mice, and whether synbiotics have a recovery effect on intestinal permeability, we detected the mRNA expression of TJ proteins in the colon. As apparent from Figure 5D, expressions of claudin3, occludin, and zonula occludin-1 were significantly reduced in ALI mice, while they increased in diet intervention groups. The results demonstrate that synbiotic treatment promoted the expression of the TJ protein.

Generally speaking, changes in intestinal permeability are accompanied by the destruction of intestinal tissues. To detect the pathological damage of the colon, we used H&E to stain the colon tissues. As shown in Figure 5E, the mucosa in the ND group was intact without inflammatory cell infiltration; the colonic mucosa in the MD group was significantly shed, crypts were destroyed, and a large number of inflammatory cells were infiltrated; the LD group and the PD group had lighter mucosal defects and fewer inflammatory cells; and the LP group was almost the same as the ND group. *L. plantarum* 1201, GOS, and synbiotic treatment can alleviate pathological damage of the colon in ALI mice, and the effect of the synbiotic was the most significant.

### 3.6. Synbiotics Inhibited the Apoptotic Signaling Pathway and the Autophagic Signaling Pathway through Inhibiting the MAPK/NF-κB Signaling Pathway in CCl_4_-Induced ALI Mice

In both the liver and colon, the mRNA level of p-ERK was elevated by the CCl_4_ challenge while that of JNK was reduced, but with the *L. plantarum* 1201, GOS, or synbiotic pretreatment, the mRNA levels of p-ERK and JNK were both similar to the control group (Figure 6A,B). Moreover, the concentrations of IKK-β and NF-κB were increased in MD, while the concentration of of IκB-α was reduced. However, in the LD, PD, and LP groups, the concentrations of these proteins returned to normal (Figure 6C). We found that the CCl_4_ challenge activated the MAPK/NF-κB signaling pathway, while *L. plantarum* 1201, GOS, or synbiotic pretreatment inhibited it. In addition, the synbiotic had the most significant effect.

Compared with the ND group, the MD group exhibited higher protein expression levels of Caspase3, BAX, and Bcl-2. The CCl_4_ challenge activated the apoptotic signaling pathway, but *L. plantarum* 1201, GOS, or synbiotic pretreatment inhibited this activation (Figure 6D–F). The change in the Bcl-2 mRNA expression level is consistent with the change in the protein expression level Figure 3F and Figure 4G).

The expression levels of Beclin1, Atg5, and LC3-II in the MD group were higher than those in the ND group. On the contrary, the expression levels of autophagic proteins in the LD, PD, and LP groups were lower than those in the MD group, and similar to those in the ND group (Figure 6G–I). These results suggest that *L. plantarum* 1201, GOS, or synbiotics may normalize the autophagy induced by CCl_4_. This beneficial effect was most significant in the LP group.

## 4. Discussion

The liver is an important organ that ensures the health of the body [39]. It is well known that ALI induced by CCl_4_ is mainly related to oxidative stress and secondary inflammation caused by oxidation [40], while *L. plantarum* has been proven to have a good antioxidant capacity [41]. Since oxidative stress is vital in the process of acute intoxication with CCl_4_ [40], we studied the mechanism by which synbiotics protect hepatocytes from these two aspects. SOD is the primary substance for scavenging free radicals in organisms [42,43], and MDA is the degradation product of plasma lipid peroxide. In the case of liver cell damage, the content of GSH in the cell decreases, and various oxidative free radicals increase [44]. When the concentration of GSH in the body is lower than the critical value, all kinds of GSH-dependent enzymes will be inactivated, and the protection of oxidative free radicals is weakened [45,46]. Base changes in the activity of thiol enzymes in liver cells lead to the degeneration and necrosis of liver cells [47]. Therefore, the decrease in GSH content is a potential early activation signal of apoptosis [48]. SOD, MDA, and GSH can all reflect the body’s oxidation/anti-oxidation status. In our research, *L. plantarum* 1201, GOS, and synbiotics drastically attenuated the AST and ALT activities in the serum, ameliorated the pathological damage of the central vein, and showed an obvious antioxidant potential.

According to previous studies, if injured, the liver can change the secretion of bile through the gut–liver axis and reduce the intestinal blood supply and peristalsis, among other factors, leading to the destruction of the intestinal mucosa and an imbalance in the intestinal flora [49]. To confirm whether the intestinal environment was dysregulated in ALI mice, we measured the intestinal flora and intestinal permeability. Our study showed a significant microbiome disruption, indicating that ALI can alter the abundance and diversity of the gut bacterial community, which may directly affect intestinal function. The relative abundances of *Lactobacillaceae* and *Christensenellaceae* were observably reduced, which was also seen in the liver-injury model [50,51]. In previous studies, the mixture of *Lactobacillus* regulated the gut microbiota, increased the number of short chain fatty acids (SCFAs), inhibited liver lipid accumulation and oxidative stress, improved the intestinal epithelial permeability, and decreased the LPS entering the portal vein, thereby inhibiting liver inflammation [52]. Similarly, in our research, we found that *L. plantarum* 1201, GOS, and synbiotic pretreatment regulated the gut microbiota, which had deteriorated as shown by the Firmicutes/Bacteroidetes ratio [14] and increased the abundance of *Lactobacillaceae*. Then, ALI mice also showed an impaired intestinal barrier, and the expression of TJs was significantly down-regulated. This was relieved after synbiotic pretreatment.

With disorders of the intestinal flora, microbiota products provide a certain concentration of pathogen-associated molecular patterns (PAMPs), such as peptidoglycan and bacterial lipopolysaccharide, which bind to Toll-like receptors and nod-like receptors on the surfaces of intestinal dendritic cells, activating NF-κB in the cytoplasm, leading to the production and secretion of inflammatory cytokines and chemokines. On the one hand, this can destroy the tight junction of the intestinal epithelium and down-regulate the intestinal barrier function. On the other hand, the broken barrier can promote the entry of PAMPs into the hepatic portal circulatory system to exert biological effects and promote liver inflammation [6,53]. CCl_4_ metabolites can also stimulate Kupffer cells in the liver, release proinflammatory cytokines, and further aggravate liver injury [54,55]. Therefore, we hypothesize that the change in inflammatory cytokines may be related to the MAPK/NF-κB signaling pathway. Previous studies have demonstrated that novel selenium-glutathione-enriched probiotics (SGP) are effective in attenuating liver fibrosis by attenuating hepatic oxidative stress, inflammation, and MAPK signaling [56]. Likewise, *L. casei* Zhang reduced hepatic inflammation through modulating the TLR-MAPK-PPAR-γ signaling pathways and the intestinal microbiota in ALI mice [11]. Furthermore, Fructosetooligosaccharides (FOS) could reduce the release of nonalcoholic fatty liver disease (NAFLD) by inhibiting the p38 MAPK signaling pathway and reducing the progression of cirrhosis [57]. Our previous studies have shown that *L. plantarum* 1201 suppressed NF-κB activation in mice (unpublished). Remarkably, we found that the expression of p-ERK, JNK, IKK-β, IκB-α, and NF-κB was reduced with synbiotic pretreatment in CCl_4_-treated mice. Then, we detected inflammatory cytokines and chemokines in the liver and colon. After activation of the MAPK/NF-κB signaling pathway, the expression of most inflammatory cytokines and chemokines was up-regulated, while the expression of IL-10 was down-regulated, and the expression in LD, PD, and LP groups was similar to that in the ND group. Collectively, pretreatment with a synbiotic may regulate the balance of intestinal flora and inhibit MAPK/NF-κB signaling, thus inhibiting the secretion of inflammatory cytokines and alleviating liver damage.

Another major finding was that synbiotics mitigated apoptosis and, consequently, attenuated CCl_4_-induced ALI. CCl_4_ can aggravate apoptosis, while Polydeoxyribonucleotide inhibited hepatocyte apoptosis through the regulation of the NF-κB/MAPK signaling pathway in mice [58]. Indeed, administration of CCl_4_ contributed to an increase in the protein expression of Caspase-3, the proapoptotic factor BAX, and the antiapoptotic protein Bcl-2. BAX directly opens the mitochondrial permeability transition pore, thereby facilitating the release of cytochrome c into the cytosol and activating the apoptotic signaling pathways [59]. *L. plantarum* C88 decreased the levels of BAX and Caspase-3 and elevated the level of Bcl-2 in the liver to ameliorate AFB-induced excessive apoptosis by regulating the mitochondrial pathway and cell death receptor pathways [60]. Thus, the synbiotic may alleviate autophagy by inhibiting the MAPK/NF-κB signaling pathway.

It is known that there are some similar pathways of cell necrosis, apoptosis, and autophagy and there is significant crosstalk between these different forms of cell death [61,62]. Autophagy is critical in the regulation of cell survival/death, and more and more reports show that autophagy can worsen liver injury [63]. We speculate that *L. plantarum* 1201, GOS, and synbiotics can alleviate CCl_4_-induced liver injury through autophagy. To test this hypothesis, we tested the expression of Beclin1, Atg5, and LC3-II in mice. LC3 has been widely used as a marker of autophagy [64]. The content of LC3II was increased after acute CCl_4_ treatment for 16 h, which undoubtedly indicated that autophagy was involved in addition to the known mechanism of CCl_4_ hepatotoxicity, because the increased expression of Beclin1 and Atg5 is evidence of the autophagy pathway. Therefore, we found that the synbiotic could inhibit autophagy by inhibiting the MAPK/NF-κB signaling pathway.

## 5. Conclusions

In summary, synbiotics protect the liver and intestines through the gut–liver axis. Synbiotics protect the liver from injury by enhancing the host’s antioxidant capacity, inhibiting the MAPK/NF-κB signal pathway and the downstream Beclin1-mediated autophagy cascade, thereby reducing CCl_4_-induced autophagy (Figure 7). It is expected that the synbiotic of *L. plantarum* 1201 and GOS can be used as a new target to protect human health by regulating the balance of intestinal flora through diet. Considering the low sample size, the experimental results may have certain deviations and further studies could be optimized by expanding the sample size.

## Figures and Tables

**Figure 1 nutrients-13-04441-f001:**
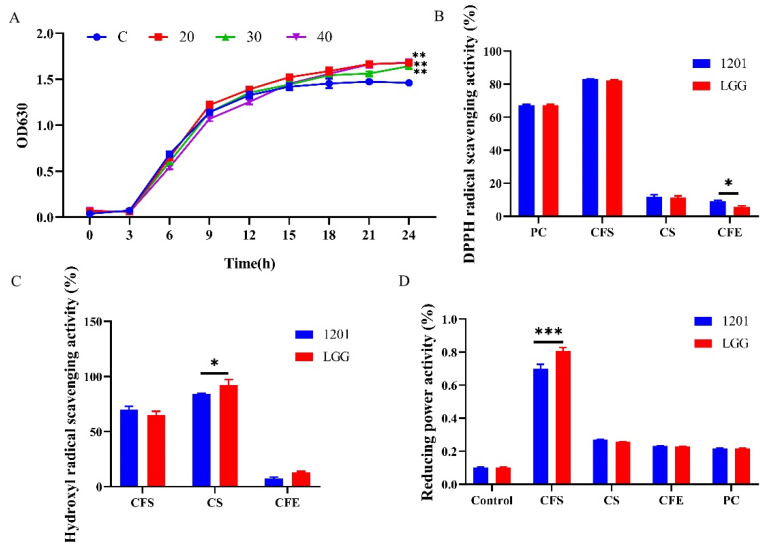
GOS promoted the growth of *L. plantarum 1201* with high-quality antioxidant power. (**A**) Growth curve of *L. plantarum 1201* in GOS treatment mediums and the MRS medium. (**B**) Picrylhydrazyl free radical (DPPH) radical scavenging assay (%) of *L. plantarum 1201*. (**C**) Hydrogen peroxide capacity (%) on *L. plantarum 1201*. (**D**) Reducing power concentration on *L. plantarum 1201*. CFS, cell-free supernatants group; CS, cell supernatants group; CFE, cell-free extracts group; PC, Positive control group. * *p* < 0.05; ** *p* < 0.01; *** *p* < 0.001; * is compared with C in Figure 1A; paired two-tailed *t*-test.

**Figure 2 nutrients-13-04441-f002:**
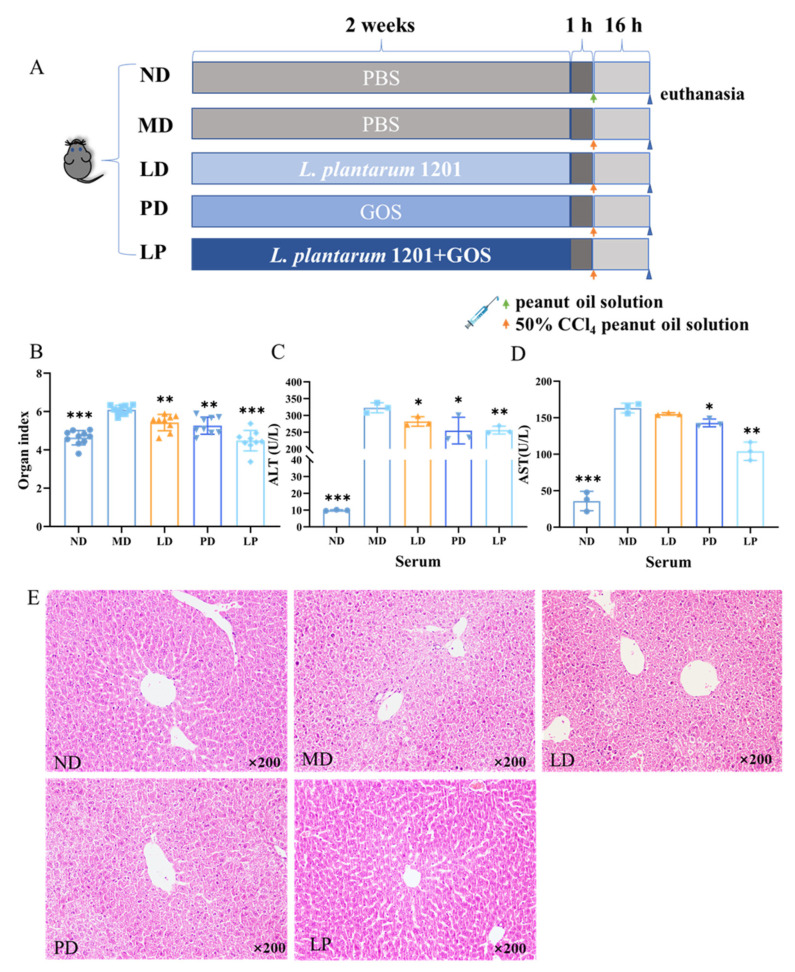
Synbiotics alleviated the dysfunction and pathological damage of the liver in CCl_4_-induced ALI mice. (**A**) Experimental set-up: C57BL/6N mice with an intraperitoneal injection of 2 μL/g 50% CCl_4_ peanut oil solution treated with probiotics, prebiotics, or synbiotics by daily gavage. (**B**) Liver index (*n* = 9 mice/group; each data point represents one mouse). (**C**,**D**) ALT and AST enzyme activity in mouse serum (*n* = 3). (**E**) H&E pathological section of mouse liver tissue (*n* = 4). * *p* < 0.05; ** *p* < 0.01; *** *p* < 0.001; * is compared with MD; paired two-tailed *t*-test.

**Figure 3 nutrients-13-04441-f003:**
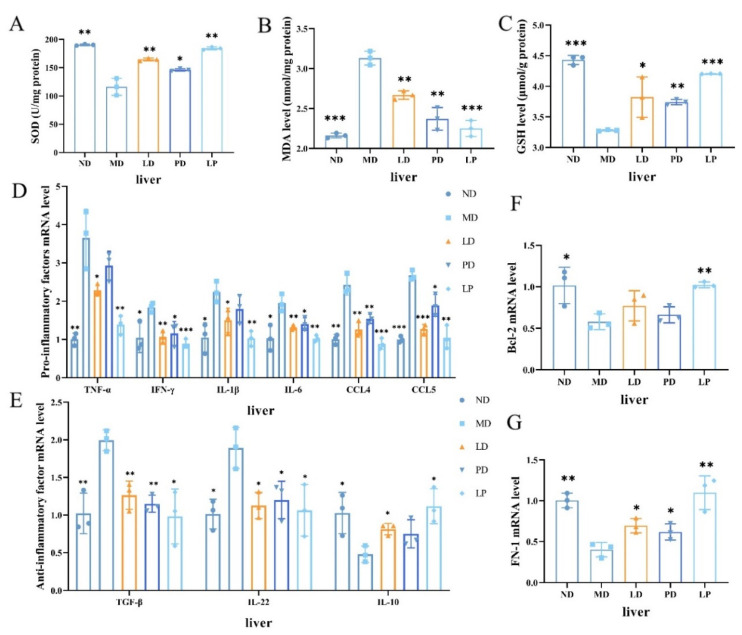
Synbiotics reduced oxidative stress and relieved liver inflammation in CCl_4_-induced ALI mice. (**A**–**C**) Homogenizing the liver tissues to examine the activities of SOD (**A**), the MDA (**B**), and GSH content (**C**) (*n* = 3). (**D**) the expression levels of IFN-γ, IL-1β, TNF-α, IL-6 and Chemokines (CCL4, CCL5) (*n* = 3). (**E**) the expression levels of anti-inflammatory factors (IL-10, IL-22, TGF-β). (**F**,**G**) the mRNA level of Bcl-2 (**F**) and FN-1 (**G**) (*n* = 3). * *p* < 0.05; ** *p* < 0.01; *** *p* < 0.001; * is compared with MD; paired two-tailed *t*-test.

**Figure 4 nutrients-13-04441-f004:**
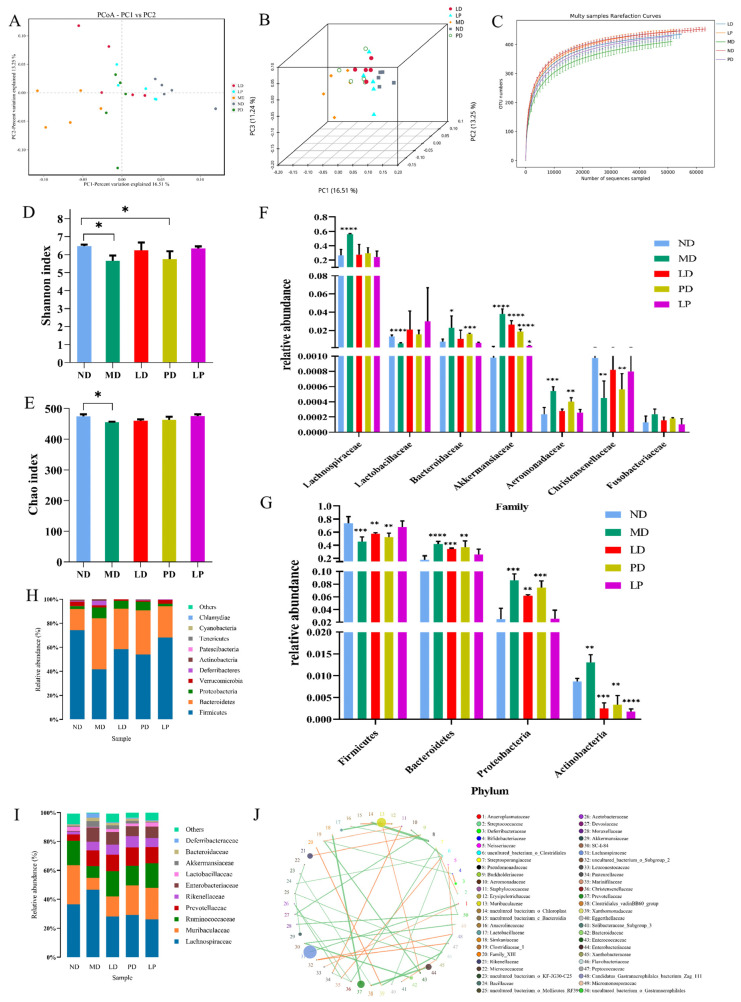
Synbiotics alleviated intestinal flora disturbance in CCl_4_-induced ALI mice. (**A**) PCoA analysis chart, PC1 vs. PC2, “the dots represent each sample; the horizontal and vertical coordinates are the two characteristic values that cause the largest differences between the samples” (*n* = 5). (**B**) Three-dimensional PCoA analysis chart, PC1 vs. PC2 vs. PC3. (**C**) Rarefaction curve (*n* = 5). (**D**) Shannon index (*n* = 5). (**E**) Chao index (*n* = 5). (**F**) Analysis of Variance between groups at the phylum level. (**G**) Analysis of Variance between groups at the family level. (**H**,**I**) Relative abundance of the gut bacterial composition at the level of the phylum and species (mean relative abundance > 0.1%). (**J**) Family-level network diagram of each species, “the circle represents the species, the size of the circle represents the average abundance of the species; the line represents the correlation between the two species, the thickness of the line represents the strength of the correlation, and the color of the line: orange represents a positive correlation, and green represents a negative correlation”. * *p* < 0.05; ** *p* < 0.01; *** *p* < 0.001; **** *p* < 0.0001; * is compared with MD; paired two-tailed *t*-test.

**Figure 5 nutrients-13-04441-f005:**
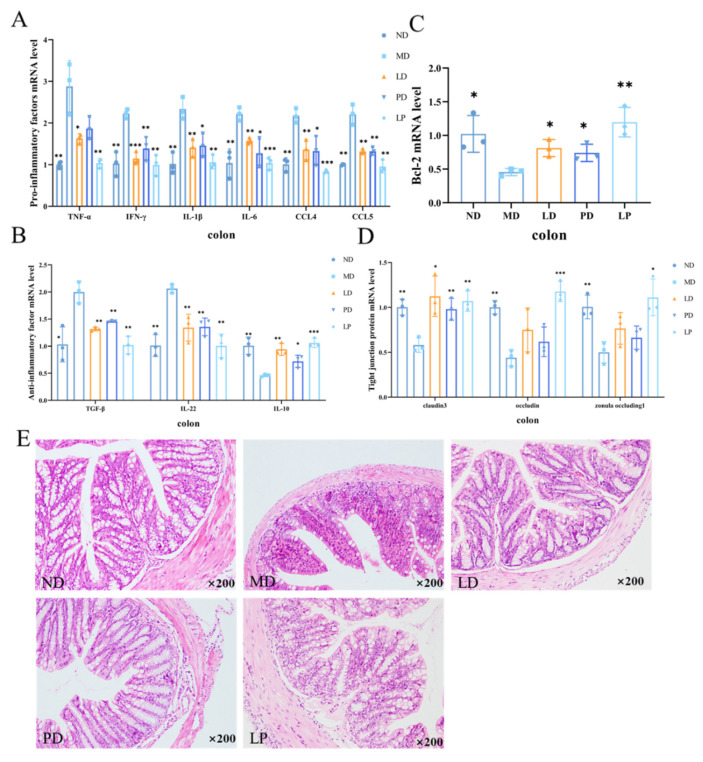
Synbiotics relieved intestinal inflammation in CCl_4_-induced ALI mice. (**A**) the expression levels of IFN-γ, IL-1β, TNF-α, IL-6, CCL4, and CCL5 in the colon tissues (*n* = 3). (**B**) the expression levels of IL-10, IL-22, and TGF-β in the colon tissues (*n* = 3). (**C**) the mRNA level of Bcl-2 (*n* = 3). (**D**) the mRNA levels of claudin3, occludin, zonula occludin-1 (*n* = 3). (**E**) H&E pathological section of the mouse colon tissue (*n* = 4). * *p* < 0.05; ** *p* < 0.01; *** *p* < 0.001; * is compared with MD; paired two-tailed *t*-test.

**Figure 6 nutrients-13-04441-f006:**
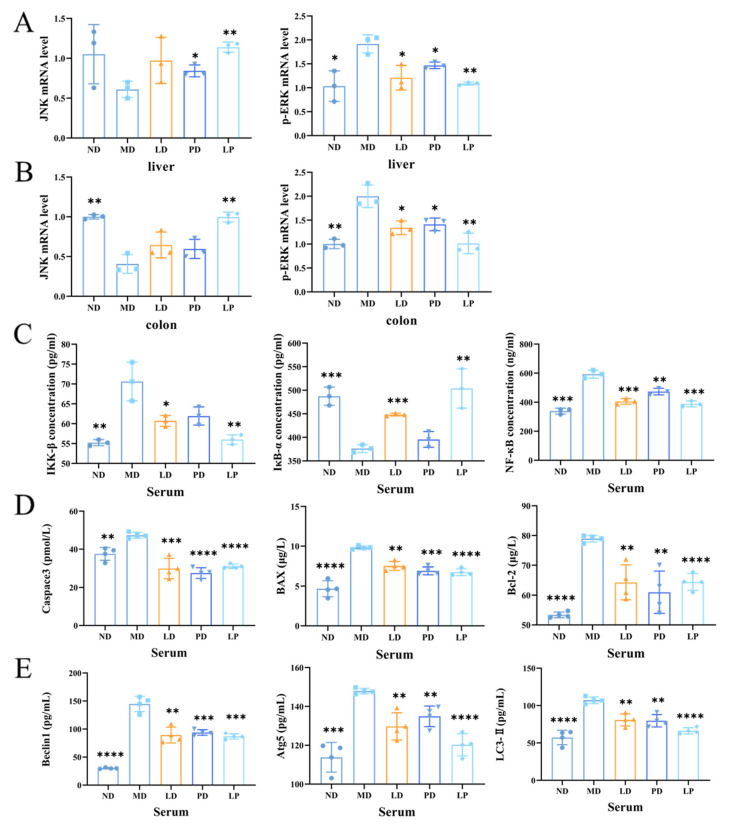
Synbiotics inhibited the MAPK/NF-κB signaling pathway, then inhibited the apoptotic signaling pathway and the autophagic signaling pathway in CCl_4_-induced ALI mice. (**A**,**B**) mRNA levels of the MAPK signaling pathway-related genes in the liver (**A**) and colon (**B**) (*n* = 3). (**C**–**E**) Quantitative analysis of protein expressions of NF-κB signaling (**C**), apoptotic signaling (**D**), and autophagic signaling (**E**) pathway-related proteins (*n* = 3–4). * *p* < 0.05; ** *p* < 0.01; *** *p* < 0.001; **** *p* < 0.0001; * is compared with MD; paired two-tailed *t*-test.

**Figure 7 nutrients-13-04441-f007:**
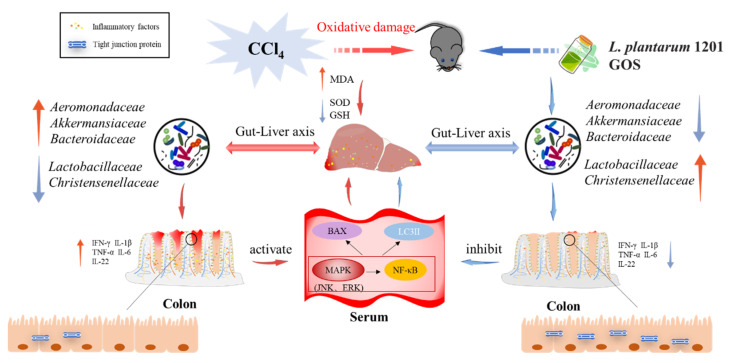
Schematic representation of the role of intestinal flora in the improvement of ALI through synbiotics. After intraperitoneal injection, CCl_4_ damages the liver, causing oxidative damage and inflammation of the liver; then it affects the balance of intestinal flora through the liver–gut axis, causing intestinal inflammation and damage, and intestinal damage in turn aggravates liver damage, causing autophagy and apoptosis. However, synbiotics act directly on the intestinal flora, relieving intestinal inflammation, inhibiting the autophagy and apoptosis pathways, and protect ing the liver from inflammation and oxidative damage.

## Data Availability

The data presented in this study are available in the article and its Appendix A.

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
