# Peer review of "Protective Effect of Lactiplantibacillus plantarum 1201 Combined with Galactooligosaccharide on Carbon Tetrachloride-Induced Acute Liver Injury in Mice"

_nutrients, 2021, doi:10.3390/nu13124441_

Round 1
Reviewer 1 Report
In this work Ren et al., study the effects of dietary intervention as possible treatment for acute liver injury (ALI). For this, the effect of Lactobacillus plantarum 1201, a probiotic with excellent antioxidant capacity, combined with galactooligosaccharide on carbon tetrachloride was studied at the microbiota level, liver and intestinal values and functions in a murine model.
The study of this probiotic as a possible treatment against acute liver injury is very interesting. However, the introduction lacks information about important aspects of the work. Although the methodology covers a lot of information, many parts are incomplete or faulty. Although the authors show many results, the explanation of some of them is insufficient or not very understandable.
Finally, the style and writing in English is very poor: Sentences are misspelled and disconnected, with several multiple bad verb conjugations.
Major comments:
- It is necessary to describe in which studies it has been shown that plantarum 1201 has a good antioxidant capacity, and if it also has an antimicrobial activity that can cause changes in the microbiome.
- In materials and methods, it appears directly LGG (Lactobacillus rhamnosus) materials and methods. It should be said what that acronym means and why it was used as a control strain in the experiment for antioxidant ability.
- Why were male mice chosen for the experiment and not both sexes, even when studies have shown that the incidence of ALI disease is very similar for both sexes or even higher in females?
- How was the gavage performed in the mice groups? According to the time of the experiment, it is not clear whether in all groups, it lasted 2 weeks or in group 3, it lasted 3 weeks. If it really lasted one more week, could this week not have affected the microbiota and the rest of the test as the animals suffered greater handling stress?
- At what point was the stool samples collected from the mice? How and with what kit was the DNA extracted? Do the authors have data only from the end of the experiment, or is there also a comparison with the beginning of the experiment to see if only the injection of peanut oil also causes a change in the microbiota of the mice?
- Authors mention fibronectin 1 (FN1) but then it doesn´t appear neither in the primers of the table nor in the results in the liver section. The same happens in the intestine with the intestinal permeability genes clan3, ocln, ZO-1. Is there a particular reason why they are named and their results have not been proven?
- How were the serum samples stored or were they tested immediately after collection? Please, describe all the processes that these samples have followed until the results were obtained.
- By what method were the contents of the colon, ileum and cecum removed? With washes with serum,….? Why were them taken if those samples have not been tested and don’t appear in the result section?
- Why has only beta diversity been studied and not alpha diversity itself within the same group? Wouldn’t be interesting to reflect the heterogeneity aspects of the community, or are they not useful for this experiment?
- Since it is clear that the disease causes damage at the level of oxidative stress, why has this technique not been studied both in the liver and in the intestine, to see how the probiotic and the oligosaccharide affect that level and is beneficial to reduce the generation of ROS?
- In the discussion, there are many aspects that have not been studied in the work itself, such as effects of the microbiota on Toll like receptors, oxidative ... If there are no results of the same situations that are not closely correlated with what found in the experiment, they should not be explained or
- Citation 35 does not exist.
Minor comments:
- The term microflora is an old term with a current use different from the one used in this paper, and nowadays is incorrect to use this word as a synonym for microbiota.
- Sentences after a period should not start with And or verbs in the infinitive form (line 201).
- The acronym GPS is written directly in the text without having explaining its meaning in the text.
- plantarum 1201 should be written in italics every time, according to the writing rules.
- Regarding the centrifugations, all should be written with g or with rpm, not with both options. And in the case of the g, they should be written all with x g .
- B- test doesn’t need the word test afterwards (line 205).
- Bax gene appears in lowercase or uppercase: The style should always be the same in the text.
- Sentence 237 uses the words slow and accelerated in the same row as if they were correlated.
- It cannot be said that both in ALT and AST the result of L. plantarum 1201 decrease significantly, because in the case of AST this is not the case. That sentence should be changed because otherwise it is misleading (line 281).
- The enzyme caspase is sometimes referred to as caspase or as caspace. Use a single and correct terminology that is appropriate for the entire text.
Author Response
Dear reviewer:
Thanks for your kindness advice for our paper (nutrients-1485103) titled as “Protective effect of Lactobacillus plantarum 1201 combined with galactooligosaccharide on carbon tetrachloride-induced acute liver injury in mice through the gut-liver axis by targeting the intestinal flora”. Here is our response for the comment from you. We would be grateful for your further evaluation, and therefore, lead to the positive reaction for accepting it for publication.
Answers to Reviewer’s comments:
(1) It is necessary to describe in which studies it has been shown that plantarum 1201 has a good antioxidant capacity, and if it also has an antimicrobial activity that can cause changes in the microbiome.
Response: Thanks for your good suggestion. L. plantarum 1201 is a strain that we screened in 2020. There is no published research on the probiotic properties of L. plantarum 1201. But in the study, we explored the antioxidant capacity of L. plantarum 1201 through three methods (line104-157, 272-290).
(2) In materials and methods, it appears directly LGG (Lactobacillus rhamnosus) materials and methods. It should be said what that acronym means and why it was used as a control strain in the experiment for antioxidant ability.
Response: Thanks for your good suggestion. we have changed in line 108-112 in the revised manuscript and the sentence is “Lacticaseibacillus rhamnosus GG has been proven by previous studies to have good antioxidant capacity[1], and some researchers have used LGG as a control to evaluate the antioxidant capacity of probiotics[2], so in this study, LGG was used as a control strain to evaluate the antioxidant capacity of L. plantarum 1201.”
(3) Why were male mice chosen for the experiment and not both sexes, even when studies have shown that the incidence of ALI disease is very similar for both sexes or even higher in females?
Response: Thanks for your good suggestion. We carefully read the previous studies when designing the experiment, and there are many animal experiments on acute liver injury using male mice[3]. At the same time, in the face of toxin invasion, such as CCl4, APAP, etc., estrogen can protect female mice[4,5]. Therefore, we choose male mice for this experiment.
(4) How was the gavage performed in the mice groups? According to the time of the experiment, it is not clear whether in all groups, it lasted 2 weeks or in group 3, it lasted 3 weeks. If it really lasted one more week, could this week not have affected the microbiota and the rest of the test as the animals suffered greater handling stress?
Response: Thanks for your good remind. After 1 week of acclimatization in the sterile room, the mice were randomly divided into 5 groups (n≥8), and they were given normal diet and water. After 2 weeks of pretreatment with L. plantarum 1201, GOS or synbiotic, CCl4 peanut oil solution was injected intraperitoneally. 16 hours after intraperitoneal injection, the mice were euthanized with ether. The gavage lasted 2 weeks, and the specific describe and graph were in line 173-190 and 318.
(5) At what point was the stool samples collected from the mice? How and with what kit was the DNA extracted? Do the authors have data only from the end of the experiment, or is there also a comparison with the beginning of the experiment to see if only the injection of peanut oil also causes a change in the microbiota of the mice?
Response: Thanks for your good remind. After euthanizing the mice with ether, we collected their colon contents and cecal contents, and sent samples of the cecal contents to Bi-omarker Tech (Beijing, China) for genomic DNA extraction, DNA purity and concentration detection, and high-throughput sequencing. We established a CCl4-induced acute liver injury model by consulting the literature[6-8], so this experiment mainly focuses on the differences between different groups, rather than the differences between the same group before and after the experiment.
(6) Authors mention fibronectin 1 (FN1) but then it doesn´t appear neither in the primers of the table nor in the results in the liver section. The same happens in the intestine with the intestinal permeability genes clan3, ocln, ZO-1. Is there a particular reason why they are named and their results have not been proven?
Response: Thanks for your good suggestion. We are so sorry for the carelessness of forgetting to put the primer sequences of FN-1 and ZO-1 into the Table S1. We have added this information in the supplementary like below. We have previously described the results of FN-1, clan3, ocln and ZO-1 in line 344-347 and 417-424.
|
FN-1 |
Forward |
ATGAGAAGCCTGGATCCCCT |
|
Reverse |
GGAAGGGTAACCAGTTGGGG |
|
|
ZO-1 |
Forward |
GATCCCTGTAAGTCACCCAGA |
|
Reverse |
CTCCCTGCTTGCACTCCTATC |
(7) How were the serum samples stored or were they tested immediately after collection? Please, describe all the processes that these samples have followed until the results were obtained.
Response: Thanks for your good suggestion. The plasma was placed in a refrigerator at 4°C for 1 night, centrifuged at 3000 rpm for 10 minutes, then the upper layer of serum was taken for storage in -20℃, and we have tested all the experiments that require serum within three days.
(8) By what method were the contents of the colon, ileum and cecum removed? With washes with serum,….? Why were them taken if those samples have not been tested and don’t appear in the result section?
Response: Thanks for your good suggestion. We cut the colon tissue and cecum tissue with sterilized scissors and forceps, and collected the contents of the colon and cecum with a sterile centrifuge tube. After washing the colon and ileum tissue with sterile 1×PBS, the colon and ileum tissue were also collected in a sterile centrifuge tube[9]. When collecting samples, I also collected ileal tissue in case there is a need for subsequent experiments to detect ileal tissue.
(9) Why has only beta diversity been studied and not alpha diversity itself within the same group? Wouldn’t be interesting to reflect the heterogeneity aspects of the community, or are they not useful for this experiment?
Response: Thanks for your kind suggestion. In this experiment, we mainly analyze differences between groups rather than differences within groups. Our aim is to explore and compare the regulating effect of L. plantarum 1201, GOS and synbiotic on the intestinal flora.
(10) Since it is clear that the disease causes damage at the level of oxidative stress, why has this technique not been studied both in the liver and in the intestine, to see how the probiotic and the oligosaccharide affect that level and is beneficial to reduce the generation of ROS?
Response: Thanks for your kind suggestion. We have tested SOD, MDA and GSH to study the effect of L. plantarum 1201, GOS, and synbiotic on liver oxidative stress in line 327-335. But we did not explore the oxidation in the intestine, because CCl4 mainly causes oxidative damage in the liver.
(11) In the discussion, there are many aspects that have not been studied in the work itself, such as effects of the microbiota on Toll like receptors, oxidative ... If there are no results of the same situations that are not closely correlated with what found in the experiment, they should not be explained or
Response: Thanks for your good suggestion. We have removed the description of the TLR4/MAPK/NF-κB pathway in the discussion and changed it to the MAPK/NF-κB pathway. Besides, oxidative is a key part of this article, so we have studied it in line 327-335 and discussed in line 481-495.
(12) Citation 35 does not exist.
Response: Thanks for your good suggestion. But after searching and confirming, we cited new literature as below.
Huang, R.; Tao, X.; Wan, C.; Li, S.; Xu, H.; Xu, F.; Shah, N.; Wei, H. In vitro probiotic characteristics of Lactobacillus plantarum ZDY 2013 and its modulatory effect on gut microbiota of mice. Journal of Dairy Science 2015, 98, 5850-5861, doi:https://doi.org/10.3168/jds.2014-9153.
(13) Minor comments:
-The term microflora is an old term with a current use different from the one used in this paper, and nowadays is incorrect to use this word as a synonym for microbiota.
-Sentences after a period should not start with And or verbs in the infinitive form (line 201).
-The acronym GPS is written directly in the text without having explaining its meaning in the text.
-plantarum 1201 should be written in italics every time, according to the writing rules.
-Regarding the centrifugations, all should be written with g or with rpm, not with both options. And in the case of the g, they should be written all with x g .
-B- test doesn’t need the word test afterwards (line 205).
-Bax gene appears in lowercase or uppercase: The style should always be the same in the text.
-Sentence 237 uses the words slow and accelerated in the same row as if they were correlated.
-It cannot be said that both in ALT and AST the result of L. plantarum 1201 decrease significantly, because in the case of AST this is not the case. That sentence should be changed because otherwise it is misleading (line 281).
-The enzyme caspase is sometimes referred to as caspase or as caspace. Use a single and correct terminology that is appropriate for the entire text.
Response: Thanks for your good suggestion. We have revised all the minor mistakes in the revised manuscript.
- Nie, P.; Wang, M.; Zhao, Y.; Liu, S.; Chen, L.; Xu, H. Protective Effect of Lactobacillus rhamnosus GG on TiO2 Nanoparticles-Induced Oxidative Stress Damage in the Liver of Young Rats. Nanomaterials 2021, 11, 803.
- Shi, Y.; Cui, X.; Gu, S.; Yan, X.; Li, R.; Xia, S.; Chen, H.; Ge, J. Antioxidative and Probiotic Activities of Lactic Acid Bacteria Isolated from Traditional Artisanal Milk Cheese from Northeast China. Probiotics and Antimicrobial Proteins 2019, 11, 1086-1099, doi:10.1007/s12602-018-9452-5.
- Chen, R.; Wang, Q.; Zhao, L.; Yang, S.; Li, Z.; Feng, Y.; Chen, J.; Ong, C.; Zhang, H. Lomatogonium Rotatum for Treatment of Acute Liver Injury in Mice: A Metabolomics Study. Metabolites 2019, 9, 227.
- Sutti, S.; Tacke, F. Liver inflammation and regeneration in drug-induced liver injury: sex matters! Clinical Science 2018, 132, 609-613, doi:10.1042/CS20171313 %J Clinical Science.
- Bizzaro, D.; Crescenzi, M.; Di Liddo, R.; Arcidiacono, D.; Cappon, A.; Bertalot, T.; Amodio, V.; Tasso, A.; Stefani, A.; Bertazzo, V.; et al. Sex-dependent differences in inflammatory responses during liver regeneration in a murine model of acute liver injury. Clinical Science 2018, 132, 255-272, doi:10.1042/CS20171260 %J Clinical Science.
- Xia, Y.; Wang, T.; Yu, S.; Liang, J.; Kuang, H. Structural characteristics and hepatoprotective potential of Aralia elata root bark polysaccharides and their effects on SCFAs produced by intestinal flora metabolism. Carbohydrate polymers 2019, 207, 256-265.
- Wang, K.; Yang, X.; Wu, Z.; Wang, H.; Li, Q.; Mei, H.; You, R.; Zhang, Y. Dendrobium officinale polysaccharide protected CCl4-induced liver fibrosis through intestinal homeostasis and the LPS-TLR4-NF-κB signaling pathway. Frontiers in pharmacology 2020, 11, 240.
- Zhang, R.; Zhao, Y.; Sun, Y.; Lu, X.; Yang, X. Isolation, characterization, and hepatoprotective effects of the raffinose family oligosaccharides from Rehmannia glutinosa Libosch. Journal of agricultural food chemistry 2013, 61, 7786-7793.
- Matsumoto, S.; Okabe, Y.; Setoyama, H.; Takayama, K.; Ohtsuka, J.; Funahashi, H.; Imaoka, A.; Okada, Y.; Umesaki, Y. Inflammatory bowel disease-like enteritis and caecitis in a senescence accelerated mouse P1/Yit strain. Gut 1998, 43, 71-78, doi:10.1136/gut.43.1.71 %J Gut.
Reviewer 2 Report
General comments
The title is too long, maybe “…through the gut-liver axis by targeting the intestinal flora” could be removed.
The methodology is messy, the purpose of the work is not clearly presented. Research hypotheses should be given.
Detailed comments
- Authors should use new names of “Lactobacillus” throughout the whole text as it was divided into 25 new genera. Follow: http://lactotax.embl.de/wuyts/lactotax/ and https://www.microbiologyresearch.org/content/journal/ijsem/10.1099/ijsem.0.004107#tab2
- Authors should use the correct term “microbiota” or “microbiome” instead of inaccurate “microflora” related to plants (flora).
- All abbreviations should be explained when used for the first time.
- The method of citation is not according to the journal demands.
- All Latin names of microorganisms (genus and species) should be written in italic, e.g. lines 55-56; 65-66 etc.
- “Bifidobacteria” should be written with a small letter with no italics.
- Give current definitions of probiotics, prebiotics and synbiotics in the Introduction.
- Some information from Introduction (from lines 60-77) should be transferred to Discussion.
- Lines 78-81 give references to previous studies. But I have a feeling that the results are presented in this manuscript…?
- Line 86: strains or strain?
- Lines 92-140: is this methodology related to experiments on animals? If so, it should be described. If no, it should be also explained. Lines 94-96 – these groups are different from the groups described in paragraph ‘2.5. Experimental groups’ ???
- Some typological errors are present through the whole manuscript, e.g. capital letters in the middle of the sentence etc.
Author Response
Dear reviewer:
Thanks for your kindness advice for our paper (nutrients-1485103) titled as “Protective effect of Lactobacillus plantarum 1201 combined with galactooligosaccharide on carbon tetrachloride-induced acute liver injury in mice through the gut-liver axis by targeting the intestinal flora”. Here is our response for the comment from you. We would be grateful for your further evaluation, and therefore, lead to the positive reaction for accepting it for publication.
Answers to Reviewer’s comments:
(1) The title is too long, maybe “…through the gut-liver axis by targeting the intestinal flora” could be removed.
Response: Thanks for your good suggestion. We have removed “through the gut-liver axis by targeting the intestinal flora” from the title in the revised manuscript.
(2) The methodology is messy, the purpose of the work is not clearly presented. Research hypotheses should be given.
Response: Thanks for your good suggestion. We put the purpose of the work in line 92-95, as “The aim of the present study was to investigate and compare the hepatoprotective effects of L. plantarum 1201, GOS and synbiotic (the combination of L. plantarum 1201 and GOS) using a CCl4-induced ALI murine model, to provide a new perspective for the liver protection.”
(3) Authors should use new names of “Lactobacillus” throughout the whole text as it was divided into 25 new genera. Follow: http://lactotax.embl.de/wuyts/lactotax/ and https://www.microbiologyresearch.org/content/journal/ijsem/10.1099/ijsem.0.004107#tab2
Response: Thanks for your good suggestion. We are so sorry for the carelessness and have checked the whole text and corrected the writing of Lactobacillus.
(4) Authors should use the correct term “microbiota” or “microbiome” instead of inaccurate “microflora” related to plants (flora).
Response: Thanks for your good suggestion. We have changed “microflora” into “microbiota” in line 518 in the revised manuscript.
(5) All abbreviations should be explained when used for the first time.
Response: Thanks for your good suggestion. We have added all the explanations of abbreviations when used them for the first time in the revised manuscript.
(6) The method of citation is not according to the journal demands.
Response: Thanks for your good suggestion. We are so sorry for the carelessness and have changed the style of the citation into MDPI by using EndNote.
(7) All Latin names of microorganisms (genus and species) should be written in italic, e.g. lines 55-56; 65-66 etc.
Response: Thanks for your good suggestion. We are so sorry it is our mistake and we have revised the mistakes.
(8) “Bifidobacteria” should be written with a small letter with no italics.
Response: Thanks for your good suggestion. We have changed the writing of “bifidobacteria” in line 78 in the revised manuscript.
(9) Give current definitions of probiotics, prebiotics and synbiotics in the Introduction.
Response: Thanks for your good suggestion. We have added the definitions of probiotics, prebiotics and synbiotic in line 63-68 in the revised manuscript, and the sentence was “Probiotics are defined viable microorganisms, sufficient amounts of which reach the intestine in an active state and thus exert positive health effects[1]. Prebiotic is a selectively fermented ingredient that allows specific changes, both in the composition and/or activity in the gastrointestinal microflora that confers benefits upon host wellbeing and health, whereas synergistic combinations of pro- and prebiotics are called synbiotics[2].”
(10) Some information from Introduction (from lines 60-77) should be transferred to Discussion.
Response: Thanks for your good suggestion. But this paragraph is an introduction to the experimental basis of this article, so it may be more appropriate to put it in the Introduction.
(11) Lines 78-81 give references to previous studies. But I have a feeling that the results are presented in this manuscript…?
Response: Thanks for your good suggestion. We are so sorry it is our mistake in language expression and we have revised the mistakes in the revised manuscript. These results were obtained in this experiment. The new sentence is “We found that L. plantarum 1201 had excellent antioxidant capacity, which meant L. plantarum 1201 might reduce oxidative damage to inhibit the ALI caused by CCl4.”
(12) Line 86: strains or strain?
Response: Thanks for your good suggestion. We have changed strains to strain in line 97 and 104 in the revised manuscript.
(13) Lines 92-140: is this methodology related to experiments on animals? If so, it should be described. If no, it should be also explained. Lines 94-96 – these groups are different from the groups described in paragraph ‘2.5. Experimental groups’ ???
Response: Thanks for your good suggestion. This part is an exploration of the in vitro antioxidant properties of L. plantarum 1201. It is not related to animal experiments, so there are different groups from the groups described in paragraph “2.5. Experimental groups.” But this methodology is the basis of this article, which is the reason why we chose L. plantarum 1201. L. plantarum 1201 has good antioxidant capacity in vitro, so it may reduce oxidation to alleviate liver oxidative damage caused by CCl4.
(14) Some typological errors are present through the whole manuscript, e.g. capital letters in the middle of the sentence etc.
Response: Thanks for your good suggestion, and we have modified our paper according to the suggestion of the editor in MDPI Author Services (https://www.mdpi.com/authors/english).
- Plaza-Diaz, J.; Ruiz-Ojeda, F.; Gil-Campos, M.; Gil, A. Mechanisms of Action of Probiotics. Advances in Nutrition 2019, 10, S49-S66, doi:10.1093/advances/nmy063 %J Advances in Nutrition.
- de Vrese, M.; Schrezenmeir, J. Probiotics, Prebiotics, and Synbiotics. In Food Biotechnology, Stahl, U., Donalies, U.E.B., Nevoigt, E., Eds.; Springer Berlin Heidelberg: Berlin, Heidelberg, 2008; pp. 1-66.
Reviewer 3 Report
The authors describe important findings regarding the protective properties of synbiotics on liver injury. Modifications are needed before re-review.
The manuscript contains English errors throughout and requires careful editing. The attached form of the manuscript shows words/sentences that need correction. Articles (a, the) are often absent. Singular/plural terms are often not in agreement.
Were animals individually housed? This affects the intestinal microbiota.
Line 162, Change to 16 hours after...
170, 171: Maintain past tense.
196: What are "available data?"
264: extracellular metabolism products?
Avoid starting a sentence with And or Then.
Significance test results are not reported for the GOS growth/antioxidant findings, nor are they shown in figure 1.
Define and explain the determination of liver/organ index.
Figure 2 footnote indicates n=4 for the histology photos but there were 5 animal groups. How can this be? Also, the footnote defines **** but this does not exist in the figure.
Figure 3 & Figure 5. Why were samples from only 3 mice per group analyzed? Calculate and report the statistical power achieved with n=3. What are the units of measure for mRNA level? As for Fig 2, no bars show the **** symbol.
Figure 6. Why were samples from only 3 mice per group analyzed? Calculate and report the statistical power achieved with n=3. What are the units of measure for mRNA level?
Line 429: The intestinal tract is the largest immune organ. It is arguable whether the liver is the most important organ for health.
Line 460: define SCFA
Line 474: define "it"

Author Response
Dear reviewer:
Thanks for your kindness advice for our paper (nutrients-1485103) titled as “Protective effect of Lactobacillus plantarum 1201 combined with galactooligosaccharide on carbon tetrachloride-induced acute liver injury in mice through the gut-liver axis by targeting the intestinal flora”. Here is our response for the comment from you. We would be grateful for your further evaluation, and therefore, lead to the positive reaction for accepting it for publication.
Answers to Reviewer’s comments:
(1) The authors describe important findings regarding the protective properties of synbiotics on liver injury. Modifications are needed before re-review.
Response: Thanks for your good suggestion. The manuscript has been carefully revised based on the comments of the three reviewers.
(2) The manuscript contains English errors throughout and requires careful editing. The attached form of the manuscript shows words/sentences that need correction. Articles (a, the) are often absent. Singular/plural terms are often not in agreement.
Response: Thanks for your good suggestion, and we have modified our paper according to the suggestion of the editor in MDPI Author Services (https://www.mdpi.com/authors/english).
(3) Were animals individually housed? This affects the intestinal microbiota.
Response: Thanks for your good suggestion. We divided the mice into 5 groups and reared them according to the groups. The mice in the same group are still kept together, with each cage no more than 5, thus ensuring small differences of the intestinal microbiota within the group.
(4) Line 162, Change to 16 hours after...
Response: Thanks for your good suggestion. We have changed the sentence as “16 hours after intraperitoneal injection, …”
(5) 170, 171: Maintain past tense.
Response: Thanks for your good suggestion. We are so sorry that it is our negligence and we have revised in the revised manuscript.
(6) 196: What are "available data?"
Response: Thanks for your good suggestion. As we described in Materials and Methods, the available date was the clean date, which was obtained by stitching, filtering and removing chimera of the original Data.
(7) 264: extracellular metabolism products?
Response: Thanks for your good suggestion. We are so sorry it is our mistake to write an uncompleted phrase. We have changed the phrase as “the secretion of extracellular metabolism products”
(8) Avoid starting a sentence with And or Then.
Response: Thanks for your good suggestion. We have edited the whole manuscript to reduce the sentence starting with And or Then.
(9) Significance test results are not reported for the GOS growth/antioxidant findings, nor are they shown in figure 1.
Response: Thanks for your good suggestion. We are so sorry it is our mistake to forget to market the significant symbols in Fig 1. We have revised it and the new figure is below:
(10) Define and explain the determination of liver/organ index.
Response: Thanks for your good suggestion. We have added the determination of liver index in line 195 as “Liver index was calculated as liver weight divided by body weight.”
(11) Figure 2 footnote indicates n=4 for the histology photos but there were 5 animal groups. How can this be? Also, the footnote defines **** but this does not exist in the figure.
Response: Thanks for your good suggestion. In Figure 2E, "n=4" is an illustration of the sample size, which means that we have selected the liver tissues of 4 mice for HE staining. **** does not appear in Figure 2, we are so sorry that it is our negligence. We have deleted **** here.
(12) Figure 3 & Figure 5. Why were samples from only 3 mice per group analyzed? Calculate and report the statistical power achieved with n=3. What are the units of measure for mRNA level? As for Fig 2, no bars show the **** symbol. Figure 6. Why were samples from only 3 mice per group analyzed? Calculate and report the statistical power achieved with n=3. What are the units of measure for mRNA level?
Response: Thanks for your good suggestion. Many previous documents also analyzed only 3 mice per group[1-3]. When the sample size reaches 3, the result is statistically significant. The mRNA level is a relative value. Gene expression was assessed by the comparative quantification method (2-∆∆CT), and the data were normalized to β-actin. **** does not appear in Figure 3 and Fig 5, we are so sorry that it is our negligence. We have deleted **** here.
(13) Line 429: The intestinal tract is the largest immune organ. It is arguable whether the liver is the most important organ for health.
Response: Thanks for your good suggestion. We are so sorry that it is our mistake of inaccurate description and we have revised in line 473. The new sentence was “The liver is an important organ to ensure the health of the body.”
(14) Line 460: define SCFA
Response: Thanks for your good suggestion. We have added the definition of SCFAs in line 507 as “short chain fatty acids (SCFAs).”
(15) Line 474: define "it"
Response: Thanks for your good suggestion. After confirm, “it” means “pathogen-associated molecular patterns (PAMPs)”.
- Zhu, M.; Yin, Y.; Ping, S.; Yu, H.; Wan, G.; Jian, X.; Li, P. Berberine promotes ischemia-induced angiogenesis in mice heart via upregulation of microRNA-29b. Clinical and Experimental Hypertension 2017, 39, 672-679, doi:10.1080/10641963.2017.1313853.
- Qian, W.; Yu, D.; Zhang, J.; Hu, Q.; Tang, C.; Liu, P.; Ye, P.; Wang, X.; Lv, Q.; Chen, M.; et al. Wogonin Attenuates Isoprenaline-Induced Myocardial Hypertrophy in Mice by Suppressing the PI3K/Akt Pathway. Frontiers in Pharmacology 2018, 9, doi:10.3389/fphar.2018.00896.
- Gong, Z.; Wang, S.; Huang, Y.; Zhao, R.; Zhu, Q.; Lin, W. Identification and validation of suitable reference genes for RT-qPCR analysis in mouse testis development. Molecular Genetics and Genomics 2014, 289, 1157-1169, doi:10.1007/s00438-014-0877-6.
Round 2
Reviewer 2 Report
I have no more comments
Author Response
Thanks a lot for your recognition and constructive comments of our paper.
Reviewer 3 Report
The authors have made significant improvements to the manuscript. Items that still need to be addressed are:
Line 119: "Like previous study" needs correction.
214: "were putted" is not correct.
226: "Basing" is not correct.
365, 433, 439: sentence needs correction.
The authors' defense of finding significant results despite low sample size is not tenable.
See: Button, K., Ioannidis, J., Mokrysz, C. et al. Power failure: why small sample size undermines the reliability of neuroscience. Nat Rev Neurosci 14, 365–376 (2013). https://doi.org/10.1038/nrn3475
The authors need to calculate the power for the analyses and report this in the discussion. Add a paragraph listing this and other limitations of the study.
Author Response
Dear reviewer:
Thanks for your kindness advice for our paper (nutrients-1485103) titled as “Protective effect of Lactobacillus plantarum 1201 combined with galactooligosaccharide on carbon tetrachloride-induced acute liver injury in mice through the gut-liver axis by targeting the intestinal flora”. Here is our response for the comment from you. We would be grateful for your further evaluation, and therefore, lead to the positive reaction for accepting it for publication.
Answers to Reviewer’s comments:
- Line 119: "Like previous study" needs correction.
Response: Thanks for your good suggestion. We have changed “Like previous study” to “According to the previous researches” in line 119.
- 214: "were putted" is not correct.
Response: Thanks for your good suggestion. We have changed “were putted” to “were put”.
- 226: "Basing" is not correct.
Response: Thanks for your good suggestion. We have changed “basing” to “based”.
- 365, 433, 439: sentence needs correction.
Response: Thanks for your good suggestion. We have corrected the mistakes. The new sentences are “The distribution of intestinal flora in the MD group was quite different from that in the ND group” (line 364-365), “L. plantarum 1201, GOS and synbiotic treatment can alleviate pathological damage of the colon in ALI mice, and the effect of the synbiotic was the most significant” (line 433-435), “(D) the mRNA levels of claudin3, occludin, zonula occludin-1 (n=3). (E) HE pathological section of mouse colon tissue (n=4)” (line 440-441).
- The authors' defense of finding significant results despite low sample size is not tenable. See: Button, K., Ioannidis, J., Mokrysz, C. et al.Power failure: why small sample size undermines the reliability of neuroscience. Nat Rev Neurosci 14, 365–376 (2013). https://doi.org/10.1038/nrn3475. The authors need to calculate the power for the analyses and report this in the discussion. Add a paragraph listing this and other limitations of the study.
Response: Thanks for your good suggestion. We have carefully read “Power failure: why small sample size undermines the reliability of neuroscience”. We are sorry that we only tested 3-4 samples due to funding and did not consider the statistical power factor. We have pointed out this shortcoming in the conclusion, and the sentence is “Considering the low sample size, the experimental results may have certain deviations and further study can be optimized by expanding the sample size.” In future experiments, we will pay attention to the factor of sample size. Thank you again for your suggestion, it is very important.